# Transformation of Seafood Side-Streams and Residuals into Valuable Products

**DOI:** 10.3390/foods12020422

**Published:** 2023-01-16

**Authors:** Shahida Anusha Siddiqui, Henning Schulte, Daniel Pleissner, Stephanie Schönfelder, Kristine Kvangarsnes, Egidijus Dauksas, Turid Rustad, Janna Cropotova, Volker Heinz, Sergiy Smetana

**Affiliations:** 1German Institute of Food Technologies (DIL e.V.), Professor-von-Klitzing-Straße 7, 49610 Quakenbrück, Germany; 2Department of Biotechnology and Sustainability, Technical University of Munich, Campus Straubing, Essigberg 3, 94315 Straubing, Germany; 3Osnabrück University of Applied Sciences, Albrechtstraße 30, 49076 Osnabrück, Germany; 4Sustainable Chemistry (Resource Efficiency), Institute of Sustainable Chemistry, Leuphana University of Lüneburg, Universitätsallee 1, C13.203, 21335 Lüneburg, Germany; 5Institute for Food and Environmental Research (ILU), Papendorfer Weg 3, 14806 Bad Belzig, Germany; 6Department of Biological Sciences Ålesund, Norwegian University of Science and Technology, Larsgårdsvegen 4, 6025 Ålesund, Norway; 7Department of Biotechnology and Food Science, Norwegian University of Science and Technology, Sem Sælandsvei 6/8, Kjemiblokk 3, 163, 7491 Trondheim, Norway

**Keywords:** recycling, seafood, side-streams, sustainability, treatment

## Abstract

Seafood processing creates enormous amounts of side-streams. This review deals with the use of seafood side-streams for transformation into valuable products and identifies suitable approaches for making use of it for different purposes. Starting at the stage of catching fish to its selling point, many of the fish parts, such as head, skin, tail, fillet cut-offs, and the viscera, are wasted. These parts are rich in proteins, enzymes, healthy fatty acids such as monounsaturated and polyunsaturated ones, gelatin, and collagen. The valuable biochemical composition makes it worth discussing paths through which seafood side-streams can be turned into valuable products. Drawbacks, as well as challenges of different aquacultures, demonstrate the importance of using the various side-streams to produce valuable compounds to improve economic performance efficiency and sustainability of aquaculture. In this review, conventional and novel utilization approaches, as well as a combination of both, have been identified, which will lead to the development of sustainable production chains and the emergence of new bio-based products in the future.

## 1. Introduction

The growing global population is expected to reach 9.7 billion by 2050, resulting in the search for a healthy lifestyle and appropriate meals to retain good health [1,2], as well as resulting in increasing demand for a nutritious and sustainable food supply [3]. At the same time, the production of animal foodstuffs and factory farming are associated with major negative impacts on the environment and the human health [4,5]. These reasons lead to the search for sustainable food alternatives focusing on the preservation of wasted nutrients within the food chain and conformity with the sustainable circular economy principles [6,7]. The added value given to by-products allows waste reduction and recycling to conserve natural resources, protect the environment, and focus on consumer health [7]. Nutrients and bioactive compounds present in food by-products can be an excellent source for developing new food products to improve the health and wellbeing of consumers and protect the environment at the same time [8].

Fish is regarded as a valuable component of a nutritious diet since it has a high protein content, a steady essential amino acid profile, a beneficial quantity of fat-soluble vitamins like A or D, and the necessary macro- and microminerals [9,10]. Furthermore, oily fish have considerable amounts of long-chain highly unsaturated n-3 fatty acids, which are connected with better cardiovascular health [11,12]. The fish industry has grown continuously over the past decades and rose to about 178 metric tons in 2018 [13]. The per capita intake of fish has also risen to 20.5 kg in a year in 2018 [14]. A resulting increase of fish processing comes with an increase in the quantity of fish side products, which is divided between quickly degradable items with a high enzyme content like viscera, as well as more stable products (skin, heads, and bones) [3,15,16], constituting up to 60% of the fresh fish. As the demand for marine crustaceans grows, so will the amount of processed side-streams [3]. Therefore, there is an urgent need to find proper measures to deal with the increasing amounts of side-streams.

Many conventional techniques, such as enzymatic hydrolysis [17,18], mechanical treatment [19], and chemical extractions [9,20], are employed in refining seafood by-products to get high-value-added components. These techniques are efficient, but they need a lot of energy and may cause thermal deterioration of the target molecules [15,16]. Other extraction processes that require the use of organic solvents would potentially harm both health and nature, as well as cause the destruction of perishable compounds [21]. As a result, in recent years, industries and consumers prefer more environmentally safe processes for ingredient processing, often referred to as green technologies [10,18]. Several biotechnological applications have been utilized, which meet the criterion for sustainability and are likely to become the norm in the future [15]. Green extraction methods have been identified for the separation of high-added-value chemicals through microwave-assisted extraction (MAE) [22], ultrasound-assisted extraction (UAE) [10], supercritical fluid extraction (SFE) [10], and pulsed electric fields (PEF) [10,22]. These alternative methods offer numerous benefits like faster extraction, reduced demand for solvents as well as non-polluting solvents, and improved selectiveness [23]. This study will explore the challenges, as well as the potential opportunities, of such existing and emerging approaches. This review will further define the best approaches for the future utilization of fish waste from an economic, environmental, and social perspective.

## 2. Current Drawbacks and Challenges Related to Transformation of Seafood Side-Streams into Valuable Compounds

While capture fishery has reached a biological limit, aquaculture has been the fastest growing food production industry for the last fifty years [13]. In 2018, 82 million of the 179 million metric tons of fish (including fish, mollusks, and crustaceans) produced worldwide came from aquaculture. China is by far the largest market player, accounting for 35% of global fish production, followed by the rest of Asia with 34%. Estimations for the year 2030 foresee a further increase to 108 million tons of aquaculture production (+32%), which is expected to be 54% of all fish products [13]. For China, the dominant role of aquaculture in fish production is even more apparent. While the production ratio between fisheries and aquaculture was 74 to 26 in 1978, it had inverted by 2018, then being 23.5 to 76.5. By this, China contributes more than 60% to the global aquaculture volume [13].

The constantly increasing amounts of fish produced, and the fact that around 70% thereof undergo further processing before entering the market, results in the accumulation of large amounts (20–80%, depending on the level of processing and the fish species) of fish waste, too. Even though fish waste represents a valuable resource consisting of many high-value components such as bioactive peptides, collagen, chitin, or enzymes, it is commonly used to produce fishmeal, fertilizers, and fish oil, or serves as direct feed in aquaculture [24]. The demand for fishmeal is very pronounced in the aquaculture industry due to its importance as the main diet for different farmed species [25]. Therefore, of the 12% of fish produced that are used for non-food purposes, 18 million tons are diverted to fishmeal and -oil [13]. In 2020, the amount of fishmeal consumed by China’s aquaculture sector peaked at 2.05 million tons [26]. Due to their standing as the world’s most important importer, China accounts for 33% of the global trade each year. Its aquaculture sector underwent further intensification and there has also been a transition “from low input, multitrophic systems (e.g., traditional carp polycultures that do not require formulated feeds) to monocultures or polycultures containing high-valued species dependent on feeds” [27]. This includes a trend towards more carnivorous fish being farmed in China which will further increase the demand for fishmeal. Carnivorous species such as grouper require high amounts of fishmeal in their diet, with fishmeal contents in feed formulations up to 50%. To satisfy the needs of their aquaculture industry, China also uses large quantities of “waste fish” for the production of fishmeal, while further 3 million tons of these fish per year serve as direct feed for high-value marine aquaculture. A “waste fish” is used to describe the smaller fish with an insignificant share of the market value of the catches [27]. According to [28], juveniles of commercially relevant species make up the main proportion of China’s “trash fish” (~32–50%).

This practice puts additional pressure on wild fish populations and raises food security issues in areas like Southeast Asia and Africa, where such fish are important for human nutrition [29]. A paradox when against the background of fully or over-exploited marine resources, intensifying aquaculture is deemed a remedy for fulfilling the constantly increasing demand for seafood of a growing world population and to release pressure from wild fisheries. Even though big improvements have been made by reducing the fishmeal and fish oil content in formulated feeds, e.g., by replacing some of the fish content by plant-based ingredients or products from genetically modified microorganisms [30,31], this remains a huge challenge.

Fish losses represent another drawback for the transition of side-streams into valuable products and are recognized as huge economic and environmental concerns. On one hand there are the large amounts of discarded fish, creating literally no value at all, and on the other hand, there are fish losses in the sense of diminished quality due to spoilage or physical damage. According to FAO (2020), the yearly loss and waste is estimated to be ~35% of the global harvest in capture fishery and aquaculture. In several countries, there is a lack of necessary infrastructure, especially in terms of access to electricity, drinking water, an adequate transport network, and the possibility of refrigeration, as well as services and procedures that allow for proper treatment on board and on shore to maintain fish quality [13]. Despite technological progress and innovations, the transition of seafood side-streams into valuable products is challenged by their high perishability. For example, in the case of shrimp this is considered a major problem, as in tropical climates the material is prone to rapid bacterial deterioration [32]. This is also true for fish side-stream valorization. Due to internal enzymes or bacteria, fish biomaterial readily undergoes autolysis of proteins and auto-oxidation of lipids, processes that need to be controlled to maintain the quality of by-products. This poses major hurdles for fishing vessels. Here, more advanced equipment and technologies for capture and better handling would be required [33]. For some components such as collagen, cost-efficient and sustainable extraction methods are lacking which hinders the exploitation of the potential of that side-stream ingredient at a larger scale [34].

Another side-stream of aquacultural seafood production is organic waste, consisting mainly of feces of the farmed animal and feed remains and being released in dissolved and particulate form. Its accumulation represents a major environmental problem, leading to eutrophication and organic matter pollution, respectively [35]. For example, a marked reduction in biodiversity of the benthic ecosystem in proximity to marine salmon farms was observed [36]. A sustainable way to make use of the organic waste produced by one species is using it as feed component for another. This principle is realized in integrated multi-trophic level aquaculture (IMTA), enabling reduction of organic wastes and production of additional, valuable biomass at the same time. Modern approaches to integrated intensive aquaculture include the combined farming of, for example, fish with microalgae or other seafood [37]. Although IMTA is seen as a key towards a sustainable aquaculture, its comprehensive implementation is hampered by the complexity of managing an operation with multiple species relating to production, processing, and marketing [38]. Moreover, in some areas producers face licensing issues, as regulations for aquaculture frequently prohibit or discourage nutrient recycling and reutilization of wastes as apparent in polyculture [39]. An example of IMTA that is particularly beneficial for small farmers and local food security is integrated rice-fish cultivation. Here, rice paddy fields are stocked with fish, which largely feed on weeds and pests while simultaneously fertilizing the rice crop by their droppings. This leads to increased rice yields and provides an additional source of protein and income to farmers. Although there is long tradition of this co-farming method in Asia and especially in China [40], it is nowadays only marginally adopted. Among the main constrains to a widespread use are that many farmers are lacking education in the required skills and ambivalent policy frameworks favoring intensive rice monoculture [41].

These examples of current drawbacks as well as challenges in aquaculture demonstrate the importance of using the various side-streams to produce valuable compounds. The economic importance of the aquaculture sector and the high demand for feed contrasts with the unused disposal from the mentioned side-streams. Picking up on this relationship, future utilization of side-streams can improve the efficiency and sustainability of aquaculture.

## 3. Potential behind the Transformation of Seafood Side-Streams into Valuable Compounds

Side-streams from seafood production, processing, and consumption appear in great amounts worldwide. Currently, the management of organic side- or waste streams is switching from treatment to utilization. It is a common agreement that the material utilization of side- or waste streams should have priority over energetic use. Even though the latter is still the dominant operated approach, more and more processes have been arising aiming for an almost complete utilization of organic material. The utilization of a material, irrespective of whether it is of organic or inorganic nature, is challenged by its heterogenous composition [42]. Heterogenous composition means that more than one material or compound is present, and a separation is crucial for achieving an efficient conversion into new valuable compounds. The single fractions of aquaculture side and waste-streams with a high potential as feedstock in various utilization processes are proteins, carbohydrates, as well as lipids and polyunsaturated fatty acid.

Treatment as an approach aims to minimize the risks that come along with its uncontrolled decomposition and the resulting greenhouse gas emissions. Conventional treatment processes are for instance composting, anaerobic digestion, or incineration. Composting refers to a material use as the produced compost can be applied as fertilizer on arable land. Anaerobic digestion results in the formation of methane, where incineration gives energy and heat. Furthermore, digestate, the remaining material after anaerobic digestion, can also be applied as fertilizer. Contrarily, incineration allows predominantly an energetic use. The three mentioned processes allow the utilization of organic side- and waste streams. However, the understanding of material use as an approach for the wholistic use of organic material and minimization of the risks to humans and environment is still limited. 

The aquaculture sector is expanding, and with this so are the possible side-streams which preferably need to be utilized [43,44,45]. There is wastewater and sludge from fish and shellfish aquacultures. Both streams are rich in nitrogen compounds but also phosphorous [44,46]. There are further organic solid waste streams which need to be managed [47]. Generally, the utilization of side-streams from seafood processing and consumption should follow a cascade use, and the formation of food and feed should have priority over a material and finally energetic utilization. In particular, the proteins and lipids in side-streams are highly valuable [47]. However, the utilization of by-products for the formation of valuable products would make a separate collection necessary. 

Even though wastewater treatment is state-of-the-art, it still needs further development to recover phosphorous and nitrogen compounds. In this context, recirculated aquaculture systems provide the opportunity for on-site treatment of water [44]. It is well known that phosphorous is a limited element but is essential for agriculture production. Nitrogen needs to be recovered and recycled and nitrogen waste should be avoided wherever possible, not only because of the energy-intensive ammonium formation but also because of the emission of the greenhouse gas dinitrogen oxide resulting from the degradation of nitrogen-containing biomass. Arumugam et al. (2020) extracted nitrogen and phosphorous compounds from sludge derived from shrimp and fish ponds [46]. They harvested, dried, and grounded sludge, added Milli-Q water, and treated the suspension at 105 °C and 121 °C for 2 h. Using 20 g sludge powder and 200 g water concentrations between 25 and 82 mg L^−1^ for total nitrogen and between 2 and 9 mg L^−1^ for total phosphorous were obtained. The authors used the extract as nutrient in algae cultivation and concluded that sludge extract may reduce the cost of producing microalgae and improves growth and nutritional content. 

The processing of side-streams from fish processing often considers a silage onboard trawlers to avoid microbial spoilage and to use it afterwards as feed. Silage is a simple decentralized process to conserve organic material. The stabilized material can be stored over a longer period of time and is used, for instance, as feed for broiler chicken [48]. Other simple processes are, for instance, composting [49] and incineration which, however, as outlined above, do not allow a holistic use of the material. A promising strategy is maintaining the structure of organic materials. Fish scales are considered as an innovative composite biomaterial with a highly sorted microstructure, which can be used in various fields such as wastewater treatment due to its beneficial physical and chemical characteristics. Qin et al. (2022) concluded that converting “fish scales into functional materials can avoid waste of resources and achieve great commercial value” [50]. For instance, Eswaran et al. (2021) investigated fish scale waste from *Garra mullya* as material for the fabrication of a supercapacitor [51]. Fish scales were heated for 7 h at 70 °C in 5% (*w*/*v*) and the pale nanostructured hydroxyapatite precipitate was washed, dried, crushed, and again heated in the presence of 50% (*w*/*v*) NaOH at 100 °C for 2 h. The authors achieved a material with a high conductivity and good mechanical cyclic stability. Mohan et al. (2021) examined the extraction of chitin from shells of crustaceans including shrimp, crab, squilla, and lobster [52]. Shell powder of each crustacean was first demineralized using 2 M HCl for 150 min at 60 °C. Afterwards, it was deproteinized using 3 M NaOH for 120 min at 80 °C, decolorized with a mixture of chloroform, methanol, and water in a ratio of 1:2:4 (*v*/*v*/*v*), and dried at 60 °C for 24 h. The extracted chitins were in the alpha form and of low molecular weight, as well as of nanoporous and nanofiber structures. The authors concluded that the extracted chitin can be considered for various applications. Furthermore, mussel shells revealed the potential to act as catalysts in a transesterification reaction to produce biodiesel [53].

Conserving the structure of, for instance, fish scales seems to have superior advantages in terms of value generation and applications. However, as outlined above, the processes to separate chitin [52] or hydroxyapatite [51] are complex. It seems much simpler to hydrolyze fish waste and to use the hydrolysate as feed for shrimps [54]. Fish waste hydrolysate has further been used as an organic nitrogen source for *Arthrospira platensis* [55]. The authors found 12% more protein in Spirulina compared to a control when the cultivation was carried out in the presence of 0.5% FPH. Furthermore, dry cell weight and biomass productivity increased by 34 and 39%, respectively. Hydrolysis has the advantage that different kinds of fish waste and side-streams can be processed simultaneously. Other studies also focused on the use of fish waste for the cultivation of various microorganisms such as microalgae [46,56,57]. Therefore, currently wasted residuals from fish processing can be a source of valuable components (direct extraction) or can be utilized for further cultivation of organisms, utilizing them as nutrient source. 

## 4. Traditional Methods of Transformation of Seafood Side-Streams and Residuals

Seafood side-streams and residuals are usually regarded as the material left after processing. These can include heads, backbones, viscera, skin, and cut-off in the case of fish [58] or shell and other rest raw material in case of crustaceans. By-catch is often also evaluated as valued residuals [59]. Improving fish processing technology and sanitation standards for better reuse of fisheries’ by-products is consistent with the UN Sustainable Development Goals and facilitates the exploration of new opportunities for sustainable use of marine by-products [60]. As a result, more seafood side-streams are currently processed by traditional methods of transformation into animal and fish feed, which also includes products for direct human consumption as well as food ingredients, nutraceuticals, and pharmaceuticals [59]. 

Different approaches are applied to deal with the fish production residuals (Table 1). 

The comparison of the different technologies shows that the conventional technologies are associated with various disadvantages and challenges, such as the generation of off-flavors or the negative influence on the properties of the product. For these reasons, innovative approaches are being explored to circumvent these drawbacks and consequently provide an efficient approach to the treatment of fish waste. In this context, the use of pulsed electric fields poses the greatest challenge since an application on an industrial scale is not feasible with the existing equipment. In contrast, the use of high hydrostatic pressure and ultrasound are promising approaches with few disadvantages and many advantages.

### 4.1. Enzymatic Hydrolysis

Seafood and residuals have a beneficial nutritional value. Enzymatic hydrolysis is a process that can be used to extract these proteins. Enzymatic hydrolysis is based on the cleavage of peptide bonds in proteins with simultaneous incorporation of water [90]. These reactions result in smaller peptides and free amino acids that are more water soluble compared to the original protein. The peptide obtained by using enzymatic hydrolysis are of short sequences of two to twenty amino acids [20,91]. 

Fish residuals have a high potential for the production of valuable hydrolyzed products. Today, most of the fish processing side-streams are used to produce products with low market value [3]. Peptides obtained by enzymatic hydrolysis have been shown to have multiple bioactives, like anti-oxidation or antimicrobial effects [20,91,92]. These properties are not active when the peptides are enclosed in proteins. 

In addition to proteins, seafood residuals may also have a high content of lipids [66]. Enzymatic hydrolysis can separate lipids from proteins in a mild and reproducible manner, so this method has already been investigated as an alternative for fish oil extraction [93,94]. This application can avoid organic solvents that are often needed for the extraction of oil, and since only moderate heating is required, the nutritional components of the oil can be preserved. Slizyte et al. [66] suggested that oil should be separated from the raw material prior to hydrolysis since an increased oil quality and an increased productivity of the hydrolysis could be obtained. Lipids that are present in the protein hydrolysates may result in a darkening of the fish protein hydrolysates (FPH); therefore, the removal of fat from a fatty fish is often required. The amount of oil is related to the part of fish as well as to the amount, and the viscera usually has about 10% more lipids compared to heads and frames [95]. 

A high number of studies have been carried out on the properties and potential uses of fish side-streams and by-products during the last decade (Table 1). Studies in the latest years include Sockeye heads [96]; Atlantic cod backbones; Atlantic salmon backbones/frames [66,97]; salmon viscera [62,98]; Atlantic salmon heads, frames, and viscera [66,99,100]; rainbow trout heads [101]; rainbow trout byproducts [102]; mackerel heads, frames, and viscera [103]; fish waste [104]; heads, skins, and bones from fish discards [17]; heads and bone frames from catfish [105]; viscera from red tilapia [106]; squid byproducts [107]; and frame and head from tilapia [108]. 

Enzymatic hydrolysis can be performed using endogenous or exogenous enzymes. Endogenous are naturally present in the raw material, and the production of fish silage is an example of enzymatic hydrolysis performed by these enzymes [103]. Enzymatic hydrolysis by proteases present in the digestive system of the fish was audited by [109]. Although their usage is an economical choice, the standardization of the procedure is difficult due to seasonal factors, type and number of enzymes, fish species, and by-product fraction. In contrast, exogenous enzymes are more suitable for producing food-grade protein hydrolysates [103], since these enzymes are reproducible. The proteases have specific pH levels that they require. Adjusting the pH can lead to high salt contents in the hydrolysates [103]. 

The use of exogenous enzymes can also increase the price range. Šližytė et al. [66] also showed that using exogenous enzymes relates to higher oil recovery. Exogenous enzymes can be derived from animals, plants, and microbes [20]. The endogenous enzymes are inactivated prior to enzymatic hydrolysis in some studies due to undesirable modes of action [99]. Lapeña et al. [62] conducted a study with salmon viscera, showing that endogenous enzymes are significant in increasing the yield of extracted ingredients during enzymatic hydrolysis.

Together with the optimal temperature and pH range, other factors are also important for the optimization of the enzymatic hydrolysis. Raw materials are often mixed with water prior to enzymatic hydrolysis. However, water is not always used (e.g., when viscera is subjected to enzymatic hydrolysis) and increases the processing costs linked to the drying process [98]. However, a proper dilution can maximize the product yield [103].

For industrial-scale oil extraction, unlike laboratory-scale where addition may reduce oil yield and quality, water might be required to mix the materials. The addition of water to the raw material has been shown to reduce the oil recovery due to the facilitation of the emulsion layer [66,99]. However, [66] also observed the lowest oil yield when water is added. This was likely due to oil entrapped in the viscous mixture, hindering separation of the oil. Kvangarsnes et al. (2021) observed only trace amounts of emulsion when performing a lab-scale hydrolysis of trout heads with the addition of water to the raw material at a 1:1 ratio [101]. 

The degree of hydrolysis is commonly used to quantify the progress of the reaction. To be able to evaluate the reaction in terms of its rate of progress, the degree of hydrolysis is often used in practice. This degree refers more precisely to the quotient of the number of released peptide bonds and existing bonds in the native protein. Araujo et al. [104] investigated the relationship between the degree of hydrolysis and the recovery of ingredients. A positive relationship has been observed for the recovery of proteins and oil, while the inverse linear relationship was observed between the degree of hydrolysis and collagen. 

The maritime taste, as well as the associated flavors that result from hydrolysis and a release of small hydrophobic peptides of less than 10 amino acids, pose a major hurdle in terms of acceptance process [61,103]. Aspevik et al. [103] also investigated the sensory attributes in hydrolysates from different rest raw materials prepared under comparable process conditions. In this case, the challenges could be addressed either by masking the bitter taste or by removing these peptides [61]. Recent studies demonstrated that the choice of enzyme together with the processing conditions may reduce this bitterness [66,103]. Petrova et al. [110] compared the bitterness of hydrolysates produced by different enzymes, with a combination of papain and bromelain resulting in the least bitterness. Slizyte et al. [66] found that a mixture of papain and bromelain initially increased the bitterness of the hydrolysates, but then decreased in the further course after 60 min. Different residuals from fish will contain other compounds that can influence the bitterness of protein hydrolysates. In addition, protein hydrolysates will also contain other tastes and flavors, and compounds like trimethylamine oxide and biogenic amines [103,111].

### 4.2. Thermal Treatment

Thermal treatments have been used for centuries as one of the oldest food preservation and processing methods [112]. The main purpose of this method is to inactivate pathogenic microorganisms and endogenous enzymes to ensure food safety and modify the texture, composition, and color of foods to make them acceptable for consumers, including an increase in digestibility and shelf life [113]. Today, traditional heat treatments [114] have been partially replaced by advanced processing methods such as microwave treatment [115], ohmic heating [116] and infrared heating [70].

#### 4.2.1. Conventional Thermal Heating Techniques

Conventional thermal treatment results in denaturation of proteins, including their aggregation and coagulation, which affect the quality of extracted lipids and proteins. At the same time, conventional treatment is heterogeneous and may result in yield and quality variations of the recovered ingredients. Since seafood products are highly sensitive to thermal treatments, overheating can affect their nutritional value, bioactive properties and sensory parameters (Maillard reaction products formation) [64,114]. Therefore, several research investigations were conducted to optimize the temperature and time used during the heat treatment to monitor liquid loss, sensory properties, microbiological decontamination, protein oxidation, and digestibility [111,117,118]. Traditionally, a fish oil extraction process refers to wet heat pressing method or mild cooking under vacuum applied to whole pelagic fish or cod liver in three steps: treating at high temperature (85–95 °C), pressing and centrifugation to obtain crude fish oil/cod liver oil, and refining the steps to suit edible purposes [112]. 

Normally, sardines (*Sardina pilchardus*) are processed directly into the fish oil or canned. The use of discards from the canning industry (by cooking at 95 °C for 12 min followed by pressing) gives an oil suitable as a raw fraction [119]. Processing of oily by-products such as salmon backbones usually results in high amounts of oil or proteins, but only one of them can be of high quality, depending on technology. As an alternative to conventional cooking, a two-stage processing method was proposed. The first step: thermal separation of the oil at a mild 40 °C; the second step: enzymatic hydrolysis of the remaining protein-rich fraction. As a result, up to 85% of high-quality oil fraction is separated in the first stage, and good quality fish protein hydrolysates are produced with fewer enzymes in the second stage [66]. 

#### 4.2.2. Novel Thermal Heating Techniques

##### Microwave Cooking

Microwave cooking may be used for the extraction of lipid and protein compounds from fishery side-streams [68]. This technique is characterized by the conversion of electromagnetic energy into thermal energy, and is industrially used for drying, pre-cooking, and other applications like pasteurization or microwave-assisted extraction [68]. Microwave treatment of gras carp decreased the cooking loss and maintained a compact structure of meat, while maintaining uniform salt distribution compared to traditional water bath cooking [120]. This is due to the formation of high-quality products as a result of the volumetric temperature increase and the more advantageous coagulation of proteins [68]. At the same time, microwave treatment enabled the preservation of polyunsaturated fatty acids in northern pike tissue [121]. Cooking of thermo-sensitive products such as crayfish showed that microwave treatment results in higher cooking uniformity. However, the flesh inside the tail of crayfish was more susceptible to overheating during the microwave treatment compared to conventional cooking in boiling water [67]. Microwave treatment has been used for novel microwave-assisted extraction of lipids [122] as an alternative to Bligh and Dyer [123], Soxhlet [124], and Folch et al. [125] extraction methods. Due to total rupture of the fish tissue under microwave treatment, lipids can be easier released and migrate more efficiently into the solvent. The extraction time can be reduced by 90% with a decrease in solvent consumption by 25% [122].

##### Ohmic Heating

Ohmic heating can be used for assisted extraction of lipid and protein compounds from seafood side-streams. Here, food is heated by the flow of electric current. From this follows a faster heating as well as the preservation of the nutritional value and sensory characteristics. Applications of ohmic heating are mainly limited to microbial inactivation, electroporation, inactivation of enzymes and heating of meat products [69]. However, the ability of generating pores in cell membranes opens a possibility of applying ohmic heating for mild extraction of valuable compounds from seafood raw material. A research investigation on its use for the treatment of surimi has shown that this technique may reduce the decomposition of myosin and actin, while retaining the structure and resulting in greater water retention, better color preservation, and higher concentration of sulfhydryl compounds [83]. The drawback of this method was non-uniform heating with the formation of local hot spots and cold spots [83].

##### Infrared Heating Technology

Infrared heating technology helps to extract high-value compounds from seafood rest raw materials. This technique is highly energy efficient and uses a part of the electromagnetic spectrum with wavelength range from 0.5 to 100 µm. Infrared heating causes vibrations of water molecules on a product surface and an in-depth penetration depending on the wavelength range and product properties. As a result, the product is heated, and the surface is dried. However, this treatment may increase peroxide value, due to free radical reaction, along with an increase in the number of tocopherols, due to the rupture of cell walls. Nevertheless, infrared heating inhibits the growth of bacteria, spores, yeasts, and mold and inactivate proteolytic enzymes [70]. Due to weak surface penetration, ohmic pre-cooking or combined treatment are recommended for enhanced heat treatment [71]. 

### 4.3. Extraction Techniques

#### 4.3.1. Chemical Extraction

Chemical extraction is a conventional method of extraction applying of an acid and/or alkali for recovery of valuable compounds from different food products, including seafood. Nowadays, this method is successfully applied for the extraction of collagen/gelatin, chitin and chitosan, astaxanthin, vitamins, and minerals from marine raw materials and side-streams. Collagen, as a natural protein polymer, can be found in various connective tissues. It has a wide use in different areas including cosmetic, pharmaceutical, food, and medicine industries [126]. Marine fish normally contains type 1 collagen in skin and bones [127].

The major methods for extraction of collagen/gelatin from various parts of fish species are acid solubilization and pepsin solubilization. Pepsin solubilization produces pepsin-soluble collagen and is more efficient because it leads to higher amount of collagen [127].

Fish side-streams such as skin, fins, swim bladder, and bones can be used for the preparation of type 1 collagen. Generally, in the first step of extraction, the fat is extracted by soaking the raw material in 10% butyl alcohol or another extraction solvent followed by addition of acetic acid. The relationship between the use of conventional chemical extraction and collagen yield has been studied scientifically on a number of occasions. Thus, the yield extracted from skin of Japanese seabass was 51.4%; for chub mackerel, 49.8%; for bullhead shark, 50.1%; for carp, 41.3%; for Bighead carp, 60.3%; for seabass, 15.8%; for Spanish mackerel, 13.68%; and for rohu, 78%. The chemical extraction gave a yield of collagen from the bones of skipjack tuna of 42.3%. For the recovery of collagen from fish bones of other fish, the chemical extraction gave the following yields: 40.7% for Japanese sea bass, 53.6% for ayu, 40.1% for yellow sea bream, 43.5% for horse mackerel, and 1.06% for carp. Bighead carp swim bladders produced 59% yield of collagen [128]. Under similar extraction conditions, salmon skins produced 19.6% collagen, while codfish skins gave a collagen yield of 10.9%. Normally, salmon skin is much easily solubilized under acidic conditions without a need for the re-extraction, compared to cod skin [129]. Thus, in the study of Alves et al. (2017) [129] salmon skin was completely solubilized after 72 h of acid treatment. However, the chemical extraction applied for recovery of collagen/gelatin varies from fish to fish. For example, cod skin is more resilient and needs further enzymatic extraction to recover collagen/gelatin [127].

Enzymatic treatment can be applied together with chemical extraction to assist the recovery of collagen/gelatin. Blanco et al. [126] extracted fish collagen from the skin of small-spotted catshark, blue shark, swordfish, and yellowfin tuna by using chemical and enzymatic treatments. The procedure included alkaline treatment (0.1 M NaOH for 24 h, 4 °C) followed by soaking in 10% butyl alcohol for swordfish and yellowfin tuna to remove fat. Collagen was extracted from the skin residues with 0.5 M acetic acid and 0.1% pepsin. A combination of chemical and enzymatic treatments yielded high-quality collagen in amount of 61.17% for blue shark, 33.00% for small-spotted catshark, 31.33% for yellowfin tuna, and 14.16% for swordfish. Similar treatments with acetic acid and different protease enzymes applied to bigeye tuna skin resulted in 3.05% collagen recovery. The application of trypsin and papain after acid treatment yielded soluble collagen which amounts up to 13.83% and 15.20%, respectively, while bromelain and pepsin resulted in much higher yields of soluble collagen (42.76% and 52.02%, respectively) [130]. The yield of collagen extracted from rabbitfish skin by treatment only with bromelain in a concentration of 1–2% for 2–6 h varied in the range of 3–6.5% of total skin amount, and the best treatment was found to be the use of 2% bromelain for 4 h [131].

Gelatin, as a natural biopolymer produced by thermal acidic, alkaline, or enzymatic degradation of collagen, is generally recovered by the same chemical extraction methods as collagen. Both gelatin and collagen are widely used as functional ingredients in the food industry and medicine because of their characteristic to form thermally reversible structures [132,133]. Some of the recent examples for chemical treatment of fish skin for gelatin production include the treatment of Atlantic cod skin at 50 °C for 3 h at pH 3.0, 4.0, 5.0, 8.0, and 9.0, resulting in 51.1%, 51.2%, 55.4%, 49.3%, 49.1% gelatin yield, respectively, with protein concentration of 86.5–92.8% [134]. The by-products from Alaska pollock and Pacific cod, treated by crab hepatopancreases or animal-derived proteinases for 4 h at 40 °C, produced 18% ± 2% dry fish gelatin powder [132].

Chemical extraction has also been used for the recovery of chitin and chitosan from crustaceans. Chitin is a polysaccharide obtained from shrimp and crab shells or fish scales. Chitin is utilized to produce a vast array of its oligomers as chitosan which is obtained by hydrolytic deacetylation of chitin [59]. The traditional methods for the recovery of chitin from shells of crustaceans are linked with issues such as high energy demand and hazards for the environment. It includes three steps: deproteinization with alkali treatment at high temperatures, demineralization using hydrochloric acid, and bleaching/discoloration of the shell pigments [72]. With the help of alkaline proteases, it is possible to avoid deproteinization with alkali treatment of blue crab and shrimp shell waste and reach 85–93% deproteinization degree [135,136]. Lactic fermentation of *Allopetrolisthes punctatus* crabs for 60 h excludes chemical treatment and produces 92% yield of chitin compared to traditional chemical treatments [137].

Chemical treatments can also be applied for recovery of antioxidants from fish raw materials. One of the recent examples includes the treatment of fish side-streams with HCl (0.2–1.0 M) to demineralize raw materials and further deproteinize it with 0.1 M NaOH. These treatment steps resulted in increased antioxidant activities of the material. Furthermore, ethanol extraction was used to collect antioxidant-containing components from the treated material [138].

#### 4.3.2. Supercritical Fluid Extraction

Alternative, greener, and more sustainable extraction might be achieved using supercritical fluids. For instance, collagen can be applied using CO_2_ acidified water, as it was shown in the study of [73] on collagen recovery from salt brine Atlantic cod skin. The investigation has shown that the water acidification by pressurized CO_2_ at 50 bar and 37 °C for 3 h results in 13.8% yield of collagen.

## 5. Innovative Technological Pre-Treatments for Enhanced Extraction of Valuable Compounds from Seafood Side-Streams and Their Sensory Attributes

Various extraction processes have been developed so far to recover valuable compounds from marine raw materials, including enzymatic hydrolysis, chemical-assisted extraction, pressing and cooking under vacuum, etc. The main parameters to be considered during extraction processes are yield, safety, and quality of the obtained compounds. However, traditional methods of extraction listed above may result in degradation of bioactive compounds such as enzymes, thermolabile vitamins and polyphenols, as well as oxidation of polyunsaturated fatty acids [113]. At the same time, conventional processing and extraction approaches may affect sensory quality of recovered compounds such as color, taste, bitterness, and texture due to structural and conformational changes in food molecules. Therefore, to improve the quality and increase the yield of valuable compounds recovered from a wide range of raw materials, including seafood side-streams, food professionals should constantly look for more advanced treatments and adapt new innovative processing technologies.

The constantly increasing market pressure for novel attractive ingredients with high bioactive and nutritional properties resulted in the emergence, further development, and use of non-thermal approaches, which exert minimal or no effect on the preservation of essential nutrients and sensory characteristics of food ingredients [81,88,139]. These advanced approaches have a potential to partially, or completely, replace the well-known and largely used conventional methods of extraction [140]. Non-thermal approaches are widely applied for extraction of valuable compounds from different raw materials including fruits, vegetables, seeds, meat, poultry, and seafood due to their ability to inhibit the activity of certain microorganisms and destroy cell walls of food matrices to enhance the release of bioactive compounds without destroying their bioactivity and nutritional profile [141]. The inhibition mechanism of non-thermal approaches is a key factor for the replacement of conventionally used thermal inactivation of enzymes during enzymatic hydrolysis of seafood raw material.

### 5.1. High Hydrostatic Pressure

High hydrostatic pressure (HHP) is a non-thermal, cold pasteurization technique involving the use of a liquid (normally water) as a medium for transferring the desired pressure to a product in a temperature range from 0 °C to 90 °C. A food product is sealed in its final packaging and further submerged in cold or room-temperature water within an enclosed vessel. The product is then subjected to hydrostatic pressure treatment (normally from 200 to 900 MPa) transmitted by the water. While HHP treatment involves continuous and rapid pressurization of the product without gradient and at low temperatures, the comparison between this and thermal processes is often found in the literature, where HHP seems to be more suitable to preserve the food products without affecting the product properties [74,79,142]. Currently, this approach has mostly been applied for inactivation of enzymes and microorganisms [143] in food products like meat [144,145] and fish [24]. This approach has also been widely applied for inactivation of spoilage bacteria in fruits [79,146,147] and in the juice industry [148,149,150].

However, besides the cold pasteurization effect, HHP may be used as non-thermal pre-treatment prior to enzymatic hydrolysis to assist the enzymatic hydrolysis of both plant-based and animal-based raw material, including seafood to increase the yield and functional and nutritional properties of recovered peptides [75,77]. To accelerate the hydrolysis procedure, a higher number of the binding sites of protein molecules should be exposed to the enzymatic attack. In this regard, mild HPP treatment (300–400 MPa) can be applied to induce protein unfolding [151]. High pressure leads to structural and conformational changes of proteins, which improve the efficiency of the enzymatic cleavage [75,151]. Moreover, HHP may increase the activity of certain enzymes during the hydrolysis of proteins [152]. The increased enzymatic activity and exposure of susceptible peptide bonds to enzymatic cleavage results in faster proteolysis and reduced duration of hydrolysis [153]. Other advantages of HHP over enzymatic hydrolysis include better protein digestibility and antioxidant activity of the resulting hydrolysates [75].

However, the beneficial effect of HHP on proteolysis may vary depending on extrinsic and intrinsic factors, such as processing conditions and type of raw material matrix (soft/hard texture, fish mince, small/big pieces, protein concentrate) [79]. An increase in yield of the desired substances during extraction by increasing the pressure to the critical value is associated with a resulting cell permeability [80,154]. Normally, the operating high hydrostatic pressure conditions are in the range of 100–1000 MPa at a temperature of 5–35 °C depending on the food product and target compounds to be extracted [80]. However, the best yield results for both animal and plant proteins were reported for mild HHP-treatments [75,77,152].

Different studies [75,77,155] analyzed the effect of HHP treatment on the yield as well as the quality of extracted compounds, suggesting that both intrinsic factors such the nature of raw material (plant-based or animal-based), structure, physical state of the food matrix, physicochemical and biochemical properties of the product, and extrinsic factors are relevant parameters to affect the final yield of the extracted compounds. Interestingly, it was also revealed that HHP treatment may increase the number of bioactive compounds extracted from both plant and animal-based tissues, thus enhancing the antioxidant activity of the recovered ingredients. This phenomenon occurs due to pressure-induced damages in the cellular matrix, enhanced mass transfer, and the release of matrix-bound bioactive compounds such as vitamins and certain enzymes, as well as increase in soluble protein content due to unfolding of protein structures resulting in higher fractionation of proteins during enzymatic proteolysis and generating smaller peptides with high antioxidant activity [75,77,156,157,158,159].

During the past 30 years, from the position of an emerging processing method, HHP has transformed into an industrially reliable and commercially available option in many countries. Thus, it can be successfully used at the seafood processing companies to enhance enzymatic hydrolysis of marine raw material and optimize the amount and the quality of recovered protein hydrolysates.

### 5.2. Pulsed Electric Field

Pulsed electric field (PEF) is another non-thermal approach, which is used to assist extraction of precious protein compounds from marine raw material. Today, PEF is mainly used to soften plant tissues, especially in potato processing, such as in the french fries industry, to ensure safety, high quality, and nutritional value, as well as to increase the shelf life [81,160]. A typical PEF treatment involves the application of short electric pulses (1–100 μs) in a wide range of electric field intensities (low-, moderate-, and high-field intensity). This treatment leads to reversible and irreversible permeabilization of cell membranes [81,88,161,162]. Permeabilization of plant cells is normally reversible and occurs under low PEF intensities (0.1–1 kV cm^−1^), resulting in the release of intracellular compounds through a generated, temporary permeability of the cell membrane. This procedure is currently applied to enhance the extractability in the processing of different agri-food raw materials, but may also be applied for extraction of thermolabile compounds from animal-based matrices [80,81,82,84,163].

Moderate intensities (1–5 kV cm^−1^) result in irreversible permeabilization of both plant and animal cells, while high intensities (15–70 kV cm^−1^) lead to the same effect for microbial cells [81]. Thus, the application of high PEF intensities may help to inactivate or inhibit proteolytic and degradative enzymes in seafood raw materials prior to enzymatic hydrolysis, as well as spoilage bacteria and other microorganisms present in seafood, thereby providing safety and neutralizing endogenous enzymes prior to hydrolysis for the controlled extraction of proteins [164]. Moreover, the PEF technique, as a cold pasteurization approach, is considered a reliable emerging approach able to ensure a significant microbial inactivation in liquid and semi-liquid food matrices with a minor impact on nutritional value, physicochemical quality parameters, and a number of health-beneficial compounds. Despite the fact that the intensity for microbial decontamination purposes may reach values equal or above kV cm^−1^ with a total energy supplied to the product of 40–100 kJ L^−1^, the product temperature can be kept below 40 °C [165]. Therefore, one of the main benefits of PEF application for extraction of valuable protein compounds from seafood raw material is a very short exposure time to a pulsed electric field, eliminating the chance of heating. Therefore, undesirable transformations in the food matrix interrelated with high temperatures (oxidation, destruction of vitamins, and protein aggregation, etc.), are eliminated [166]. However, electroporation may still induce the oxidation of lipids rich in polyunsaturated fatty acids due to a free radical chain reaction mechanism, as it was shown in the study of Cropotova [167]. Fish protein hydrolysates obtained from fatty fish species, such as trout or salmon, normally contain small amounts of lipids which may further trigger protein oxidation reactions due to the influence of high electrical pulses. However, this potential disadvantage should be thoroughly studied for each separate fish species hydrolysate. Similar to the PEF-assisted recovery of high-quality compounds from plant- and animal-based raw materials based on permeability and/or rupture of cell membranes [80,81,82,84,163,168], this approach may successfully be applied for extraction of protein ingredients from seafood [164].

Fish protein hydrolysates contain small bioactive peptides (<3 KDa) with strong antioxidant activities with beneficial properties like a high nutritional value. Conventional extraction methods such as isoelectric precipitation, acid/alkaline pretreatment, or enzymatic hydrolysis can negatively affect the properties of the extracted proteins. Acid/alkaline pretreatment and enzymatic hydrolysis have been already explained above. Isoelectric precipitation can be achieved by adding acid or alkaline and shifting the pH until proteins and peptides reach their isoelectric point with reduced solubility. Contrarily, PEF is a non-thermal emerging approach which helps to avoid thermal and acid/alkaline pre-treatments during extraction of proteins from raw biomaterials [23], and can further benefit the properties by activating biological activities [169,170]. However, a high-intensity electric field could also result in aggregation of proteins [171].

Recently, the PEF technique has been applied to enhance the recovery of proteins from marine raw material. The maximum extraction yield of mussel protein achieved after the application of PEF treatment was, for example, 77.08% (*w*/*w*), which is significantly higher in comparison to traditional methods of extraction [172], while PEF-assisted enzymatic hydrolysis of abalone raw material enabled the recovery of a hydrolyzed, high-quality abalone viscera protein with high yield and beneficial emulsifying characteristics [173]. 

Thus, PEF treatment can be successfully applied as a pre-treatment method before enzymatic hydrolysis of marine raw material [170]. This was very well demonstrated in the mentioned study of Li, Lin, Chen, and Fang (2016) [173] about viscera protein. Under the optimal PEF extraction conditions (intensity strength of 20 kV cm^−1^, treatment time of 600 s, ratio of material to solvent 4:1), a fully hydrolyzed product with high yield and improved characteristics was obtained compared to conventional enzymatic extraction.

PEF-assisted extraction has proven to be a promising approach for the recovery of various compounds from seafood raw materials and by-products including chitosan, collagen, calcium, chondroitin, lipids, and proteins [174,175]. However, regardless of all the advantages of PEF treatments listed above, until now this technology remains a challenge for industrial up-scaling and commercial use due to lack of reliable equipment that could be used under industrial conditions [80,176].

### 5.3. Ultrasound

Ultrasound (US) technology (otherwise called ultrasonication) is also one of the promising non-thermal approaches for this task. “Ultrasound” represents sound waves that exceed the human audible frequency range, i.e., are greater than 20 kHz. The main principle of ultrasound is reflection and scattering of acoustic waves originated from molecular movements oscillating in a propagation medium and generating compressions and decompressions which further result in an increase in mass transfer, turbulence, and production of energy [86]. At present, US is considered an emerging technology with a great potential and number of applications in many fields [87]. Being already a well-known and well-established approach in many processing sectors in the 1990s, it has recently gained an increased interest among food professionals and consumers due to its ability to preserve quality and guarantee safety of food products without deteriorating their nutritional value and health properties, as well as to extract high value compounds from different raw materials [87,88].

High-intensity sonication (20–100 kHz, >1 W cm^−2^) waves induce acoustic cavitation due to the generation and further collapse of bigger bubbles, releasing a high amount of energy [86]. Low frequencies of ultrasound (5–10 MHz, <1 W cm^−2^) lead to unstable cavitation and the bursting of bubbles above 20 kHz. At higher frequencies (>1 MHz), however, the effect of acoustic flow becomes predominant [87]. Low-energy sonication waves are mainly used for non-destructive methods of analysis in medicine, cosmetics, and the food industry, as well as in quality control (homogenization and/or emulsification efficiency, container filling control and fluid flow) [177,178]. High intensity (from 10 to 1000 W cm^−2^) and low-frequency (from 20 to 100 kHz) ultrasound is considered disruptive due to detrimental influence on the physical (including structure and mechanical properties), physicochemical and biochemical characteristics of biological materials in contrast to low-energy ultrasonic waves. This phenomenon found a wide application in the food industry for improved emulsification and foaming operations, freezing and thawing, concentration, drying, tenderization, as well as control and modification of microstructure and textural properties of fatty and protein-rich foods [61,179,180,181]. Because of cavitation produced by the high-intensity US, this technology is also used in the seafood industry to inactivate degradative enzymes, eliminate spoilage bacteria, and optimize extraction, while reducing adverse effects [89,182]. A large number of various classes of compounds have effectively been extracted from different seafood raw materials by using US [175,183,184,185]. The ultrasound assisted extraction procedure can be explained by the mechanical break of the cell wall through the implosion of bubbles and thus facilitating the penetration of the solvent into the cells and enhancing the release of intracellular material into the medium [186,187,188]. Therefore, this approach has occupied a special niche in the seafood and food processing industry for the facilitated extraction of valuable ingredients aiming to decrease the extraction time and increase the yield of isolated compounds with less detrimental changes in the quality parameters due to lower processing temperatures [87,175,184,185,188]. This approach has become an efficient technique for industrial applications already for a decade, the equipment design for commercial large-scale use with continuous-flow systems has only been optimized recently [87].

## 6. Emerging Biotech Approaches

Aquaculture is strongly connected to agriculture due to the required feed and its production, which is responsible for 87% of aquaculture GHG emission [189]. Contrarily, as outlined above, aquaculture produces large amounts of side-streams and residuals. Thus, using a novel biotechnology approach as feed for aquacultures can be directly produced from side-streams and residuals, with aquaculture disconnected from agriculture and consequently GHG emissions minimized. The recently started European project “ClimAqua” is specifically focusing on such an approach. In “ClimAqua”, the disconnection of aqua- and agriculture is reached by case- and location-specific side-streams and residuals valorization in feed production via the cultivation of microalgae. This strategy of feed production can be case-specific and adapted to the species reared in aquaculture. The result can be an almost completely digestible feed. Furthermore, a biotechnological side-stream and residuals utilization approach can not only contribute to the use of protein materials, but also to the use of sludge and wastewater. It can be carried out decentralized (on place processing) where aquacultures are located, which minimizes transportation of side-streams, residuals, and feed. ClimAqua considers a full recirculation of aquaculture side-streams as feedstocks in algal biomass, and thus livestock feed production. Wastewater and sludge streams cannot be avoided and require treatment. Particularly, avoidance measures are often beyond the stakeholders’ capacities involved in aquaculture industries [190]. Valorizing side-streams would not only make a treatment, such as incineration, unnecessary, which alone already contributes to a reduction in GHG emission, but would allow the climate beneficial and cost-efficient production of new feed. In ClimAqua, the sludge resulting from aquaculture, consisting of residues from the animals as well as the unused feed, is hydrolyzed and fed to heterotrophic algal strains. Contrarily, phototrophic strains do not grow on hydrolytic products and require nitrate as well as phosphate present in enormous amounts in wastewater from aquaculture. Recirculation of those nutrients would minimize the environmental impact of aquaculture, minimize GHG emissions, and pave the way to resilient aquaculture-based food system by 2050 and improve conditions of an advancing climate change. ClimAqua aims for a flexible algal biomass production system not limited to certain geographic areas and climate zones. For instance, it will be tested in South Africa (moderate and subtropical climate) and Norway (temperate and marine climate). The long sunshine duration over the year in South Africa is beneficial to a phototrophic cultivation, while the relatively long dark duration in Norway indicates a heterotrophic cultivation. It is expected that making use of regional environmental conditions, such as temperature and sunshine duration, can contribute to GHG emission deduction by reduced temperature regulation and artificial illumination. The development of bioprocesses which are adapted to climate and available nutrients is an emerging approach, and necessary to cope with future challenges. 

## 7. Conclusions and Future Prospects

Different approaches to the future treatment of fish waste were investigated and compared. The different approaches have some advantages and disadvantages, which define their applicability. Since the traditional methods are associated with many disadvantages, innovative novel approaches can be a feasible alternative. Their potential benefit is based on the biomass transformation with biological means (fermentation, cultivation, hydrolysis) which avoids potential biological and chemical contamination hazards. It is expected that a combination of approaches and technologies would result in the development of sustainable production chains and emergence of new biobased products.

## Figures and Tables

**Table 1 foods-12-00422-t001:** Comparative analysis of different approaches dealing with marine product residuals.

Approach	Characteristics	Efficiencies	Limitations	Benefits	References
Hydrolysis					
Hydrolysis	Cleavage of peptide bonds in proteins with inclusion of water, resulting in production of smaller peptides and free amino acids	The yield of hydrolysis is influenced by the enzymes and residues used	Development of a bitter taste and unacceptable flavors	Obtained peptides have various advantageous bioactive properties, which are not active before treatment with enzymatic hydrolysis	[61,62]
Conventional thermal treatment techniques					
Cooking	Inactivation pathogenic microorganisms and endogenous enzymes for food safety as well as to modify properties for the benefit of consumer acceptance	Optimizing the processing of oleaginous by-products by combining them with enzymatic hydrolysis	Impairment of the quality of extracted lipids and proteins due to protein denaturation, including their aggregation and coagulation, variations in the yield and quality of extracted ingredients, impairment of nutritional, bioactive, and sensory properties due to overheating	Large knowledge base and long time for optimization due to the long existence	[63,64,65,66]
Novel thermal heating techniques					
Microwave cooking	Industrially used for drying, pre-cooking, and pasteurization of ready meals as well as tempering of meat and fish, based on converting electromagnetic energy into thermal energy	Compact structure of the meat with uniform salt distribution due to volumetric temperature rise and the more uniform coagulation of proteins	Meat inside the tail of crayfish is more susceptible to overheating during microwave treatment than during conventional boiling water cooking	Wide range of applications (e.g., drying, pre-cooking microwave-assisted extraction)	[67,68]
Ohmic heating	Heating by passing an electric current	Ability to create pores in cell membranes, gentle extraction	Applications are mainly limited to microbial inactivation, electroporation, enzyme inactivation, and heating of meat products	Faster heating, no influence on the sensory food properties as well as the nutritional value	[69]
Infrared heating technology	Heating and drying of the product due to oscillations of the water molecules on the product surface and the in-depth penetration	Ohmic pre-cooking or combined treatment recommended due to weak surface penetration for improved heat treatment	May increase peroxide levels due to reaction with free radicals and tocopherols due to cell wall breakdown	Extremely energy efficient, inhibits growth of bacteria, spores, yeasts and molds, and inactivates proteolytic enzymes	[70,71]
Extraction techniques					
Chemical extraction	Use of an acid and/or alkali to extract valuable compounds from various foods	Extraction of collagen/gelatin, chitin and chitosan, astaxanthin, vitamins, and minerals from marine raw materials and side-streams	Traditional methods for the recovery of chitin from shells of crustaceans are extremely hazardous, energy consuming, and environmentally polluting	Chemicals are applied for the extraction	[72]
Supercritical fluid extraction using CO_2_ acidified water	Use of CO_2_ acidified water to extract collagen from fish skin	Water acidification by pressurized CO_2_ at 50 bar and 37 °C for 3 h results in 13.8% yield of collagen	Co-extraction of gelatin could not be excluded	Alternative, greener, and more sustainable way to extract collagen	[73]
Innovative technological pre-treatments					
High hydrostatic pressure (HHP)	Use of a liquid (usually water) as the medium, to apply the desired uniform pressure to a product	Inactivation of enzymes and spoilage microorganisms such as yeasts, molds, and gram-positive and gram-negative bacteria, industrially reliable technology that is commercially available in many countries	Positive effect on proteolysis may vary depending on extrinsic and intrinsic factors	Continuous and rapid pressurization of the product without gradient and at low temperatures, used as a cold pasteurization or non-thermal pre-treatment prior to enzymatic hydrolysis with several positive effects	[74,75,76,77,78,79]
Pulsed electric field (PEF)	Application of short duration electric pulses (1–100 μs) in a wide range of electric field strengths for a very short period (from nanoseconds to milliseconds)	Improves extractability, extraction of thermolabile compounds from animal matrices	Exposure to high electrical pulses may trigger further protein oxidation reactions in fish species hydrolysates, a strong electric field could destroy the intra- and intermolecular electrostatic interactions of certain peptides, challenge for industrial development and commercial deployment due to the lack of reliable industrial equipment	Significant microbial inactivation with little impact on the nutritional value, physicochemical quality parameters and the number of health-promoting compounds due to the low treatment temperature, very short exposure time	[80,81,82,83,84,85]
Ultrasound (US)	Reflection and scattering of acoustic waves, leading to increased mass transfer, turbulence, and energy generation	Great potential and a variety of applications in many fields due to its ability to produce permanent mechanical, chemical, and biochemical changes in fluids and gases	Plant design for large-scale commercial use with continuous flow systems has only recently been optimized	Maintaining the quality of food, ensuring its safety without compromising its nutritional value and health properties, inactivating degradative enzymes, eliminating spoilage-causing bacteria, facilitating the extraction of valuable ingredients with shorter extraction times and higher yields	[86,87,88,89]

## Data Availability

Data is contained within the article.

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
