# Peer review of "Transformation of Seafood Side-Streams and Residuals into Valuable Products"

_foods, 2023, doi:10.3390/foods12020422_

Round 1

Reviewer 1 Report

This review makes a detailed introduction of the development of seafood processing approaches, and the usage of these techniques for the transformation of seafood into valuable products, and identifies suitable approaches for processing seafood according to different purposes.

In my opinion, there are still some aspects need to be improved.

1.      In section 4, thermal and mechanical treatment are combined, due to both have no close relationship with each other, therefore, it is recommended to separate thermal and mechanical treatment into two independent parts.

2.      The logic and affiliation of the processing methods are a little confusing, such as microwave cooking belongs to conventional thermal treatment, but in Table 1, we cannot see their affiliation relationship, please revised this question. Furthermore, the contents summarized in Table 1 are not logically related to 4.2, because conventional thermal treatment and other heating treatments are described together in 4.2.1 (conventional thermal heating). I think Infrared heating technology, and microwave cooking belong to novel heating techniques, please check. It is recommended to list Section 4.2 as a separate section and introduce different heating treatments according to Table 1.

3.      There is a “supercritical fluid extraction using acidified water”, which was not introduced in the main context, while all other methods were introduced, so it recommends that to make a brief introduction of this method in the main context, as well as other methods.

4.      The "conclusion and future prospects " may not well reflect the content of this review. It is suggested to specifically summarize and point out several popular research directions.

Author Response

Thank you for your valuable comments. We considered all suggestions in the revised version.

Reviewer 2 Report

Line 53: Please specify that only oily fish (and not lean fish) have considerable amounts of long-chain highly unsaturated n-3 fatty acids.

Line 86: Please add the reference after the word “aquaculture”.

The authors are kindly asked to elaborate a little bit more on these areas:

Silage process as a traditional method of processing fish waste. The authors can add some recent information regarding the acid and microbial (lactic acid bacteria) silage/hydrolysis.

Also please add some information regarding the isoelectric solubilisation/precipitation process (pH-shift method) as a possible fish waste processing method to recover protein isolates.

Author Response

Thank you for your valuable comments. We considered all suggestion in the revised manuscript.